# Development of Evaluation Indicators for Sports Facilities for People with Disabilities Considering the Universal Design: Focusing on the Republic of Korea

**DOI:** 10.3390/healthcare10112151

**Published:** 2022-10-28

**Authors:** Eunsurk Yi, Sang-Wan Jeon, Ahra Oh

**Affiliations:** 1Department of Exercise Rehabilitation & Welfare, Gachon University, 191 Hombakmoero, Yeonsu-gu, Incheon 406-799, Korea; 2Exercise Rehabilitation Convergence Institute, Gachon University, 191 Hombakmoero, Yeonsu-gu, Incheon 406-799, Korea

**Keywords:** sports facilities for people with disabilities, universal design, sports facilities evaluation indicators

## Abstract

We developed evaluation indicators for sports facilities for people with disabilities and adopted the universal design to conduct Delphi surveys on sports facilities and sports experts. First, the range of universal design element reflection and the method of deriving the evaluation indicators were established through a literature review. Second, 21 experts conducted the first Delphi survey to select the important features of the seven principles of universal design and describe the necessary sub-factors to consider when designing sports facilities. The described elements were divided into 15 categories, and 49 sub-factors were extracted. Third, based on the evaluation of the indicators’ content, acceptance was investigated, and the survey data were analyzed through indicators of reliability and validity of the sub-factors and categories. Fourth, we discussed whether to accept the standard value on the basis of the evaluation index through an expert meeting. Subsequently, the final evaluation index was obtained. The developed evaluation index should be applied by the operators and users of public sports facilities, and validation work is needed. Guidelines for applying the universal design to various sports facilities for people with disabilities should be developed. The financing of sports facilities applying the universal design and related policies should be discussed.

## 1. Introduction

Currently, in Korea, aging-related problems are increasing owing to the rapid increase in the elderly population along with the economic development and advances in medical technology [1]. Specifically, the number of older people with disabilities caused by aging is steadily increasing owing to adverse environmental factors and industrial accidents [2]. This has become a social problem, and practical alternatives for the normal life of older adults and all people with disabilities are greatly required. Therefore, the government, academics, and social groups are actively investigating and attempting to meet the requirements of the elderly and disabled population, e.g., by improving the living environment of people who are underprivileged and their movement and access to facilities and information, eliminating inconveniences and obstacles.

So far, these social efforts have mainly considered specific disadvantaged groups, such as older adults and all categories of people with disabilities. “Barrier-free” is a social movement and policy implemented to remove physical and psychological barriers that hinder people who are socially disadvantaged, such as those with disabilities. Currently, people with and without disabilities live together. Therefore, as barrier-free gives priority to those with disabilities, in reality, it has caused problems by consciously alienating them, further emphasizing their disability [3]. Consequently, designing and providing separate environments and products that support the status and needs of specific users has become valuable. However, considering potential users such as the elderly and all categories of people with disabilities, a universal design (UD)—which provides designed environments and products that can be used safely and conveniently by all, so that people with and without disabilities can live together [4]—has become necessary.

The UD was developed from the concept of a barrier-free living environment [5]. It is a design that does not separate people with and without disabilities and creates an accessible environment reflecting the users’ needs and consequently suitable for all [6]. The users participate, provide opinions, and reach an agreement with experts in each field [7]. The UD has already been defined and utilized, considering the environmental characteristics of different countries, in particular the United States and Europe [8]. The UD Center of North Carolina State University in the USA performed a project titled “Studies to Further the Development of UD” (no. H133A40006). During the project activities, seven UD principles (equitable use, flexibility in use, simple and intuitive use, perceptible information, tolerance for error, low physical effort, and size and space for approach and use) were developed with a focus on the built environment, products, and communication [9]. The seven UD principles guide the design process, are used to systematically evaluate proposed designs, and help designers and consumers characterize more useful design solutions [10,11].

In the United Kingdom, guidelines for sports facilities with an inclusive design, a concept similar to the UD, were presented at the national level [12]. In addition, Japan had 114 UD sports facilities as of 2014. Many European countries encourage the use of sports facilities, regardless of whether people have disabilities [13].

In contrast, in Korea, convenience facilities for people with disabilities are well established owing to legal regulations. However, sports facilities not satisfying the legal standards are not suitable for people with disabilities. A recent study investigated the demand by users and operators of public sports facilities of UD sports facilities for people with disabilities. It found that non-disabled users were favorable to the construction of UD sports facilities and the use of related programs. In addition, to become a sports facility that can be used fairly by people with and without disabilities, it is necessary to expand the scope of the legislation related to the installation of convenience facilities for people with disabilities in sports facilities [14]. Recognizing the importance of sports facilities that can be used by anyone without discrimination, the government announced plans to increase the number of sports centers to 150 by 2025 [15].

In accordance with this trend, in Korea, the environment and facilities have been evaluated through the “life environment certification system without obstacles”. However, appropriate evaluations have been difficult, owing to the burden of costs and procedures; therefore, integrated and emotional evaluations of the environment and facilities are not applied [16]. Considering the number of future public sports facilities, the development of evaluation indicators to objectively evaluate various sports-related facilities is required. Consequently, institutional directions, such as strengthening user services, convenience, and safety support, have been suggested.

Concerning research on sports facilities that apply the UD, studies have been published on the development of a user perception measurement tool [17] and the creation of a learning environment [18], reporting examples of facilities that can be understood from users’ viewpoints. Some studies only dealt with methods [19,20], and only a limited range of topics have been covered. The UD for a living environment where various users coexist has been frequently applied in the public environment and facility fields by local governments. Furthermore, it has had a great influence on related academic research fields. However, studies on sports facilities for people with disabilities that consider the UD are required. Specifically, research on this topic is necessary because no indicators have been developed to objectively evaluate sports facilities to which the UD is applied.

Therefore, we established the scope of UD elements and a method of deriving evaluation indicators through a literature review, extracted the factors to be considered for UD-applied sports facilities through two Delphi surveys, and determined the elements of the evaluation indicators through expert meetings.

## 2. Materials and Methods

This mixed-methods study utilized qualitative (expert interviews and content analysis) and quantitative (frequency analyses) instruments.

### 2.1. Participants (Panel Experts)

The participants had worked in a specialized field for more than 10 years and had an analytical perspective on the research topic and expertise with abundant field experience; thus, they could provide insight into the overall context and process. The panel consisted of 21 people from four fields: seven architects, seven experts on people with disabilities, four experts on older adults, and three UD experts. Table 1 shows the personal characteristics of the panel experts. The selection of the study participants was conducted in accordance with the guidelines of the Helsinki Declaration and was approved by the Gachon University Institutional Review Committee (no. 1044396-202007-HR-125-01).

### 2.2. Instruments and Procedure

#### 2.2.1. Procedure

A standard feasibility study was derived over four steps. First, the range of UD element reflection and the method of deriving the evaluation indicators were established through a literature review. Second, 21 experts conducted the first Delphi survey to select the features applied preferentially to the seven UD principles and described the sub-factors to be considered when designing sports facilities. The described elements were divided into 15 categories, and 50 sub-factors were extracted. Third, based on the evaluation indicators content classified through the first Delphi survey, acceptance was investigated through a five-point Likert scale, and the survey data were analyzed through indicators of reliability and validity of the sub-factors and categories. Fourth, we discussed whether to accept the standard value as a result of the evaluation index through an expert meeting. Subsequently, the final evaluation index was completed. The detailed study procedure is shown in Figure 1.

#### 2.2.2. Systematic Literature Review

The scope of UD element reflection and the method of deriving evaluation indicators were established through a systematic literature review. According to the process of literature search suggested by Petticrew and Roberts [21] and Fatorić and Seekamp [22]—“create keywords”, “conduct search”, “collect publications”, and “select publications”—the literature search was conducted in the order of “analyze publications” and “report and discuss”.

The keywords were “UD facilities” and “sports facilities”, and the Korean literature was searched using the “Research Information Sharing Service” provided by the Korea Education and Research Information Service. The international literature was also searched using “Google Scholar”. The publication period of the literature was set from 2011 to 2022, and bibliographic information was included, on the basis of the following steps. First, duplicate data were sorted, and selection and exclusion criteria were applied based on the article titles. Second, selection and exclusion criteria were applied based on the abstracts. When it was difficult to select a document only based on the abstract, the full text was searched and confirmed. Only data from studies or reports from public institutions were considered. Through this process, 80 domestic studies and 237 international studies were primarily searched. Third, the studies for the final analysis were selected through the title and abstract. The subjects of the selected literature were “UD facilities”, and the subjects of interest were “evaluation” and “guidelines”. The context of the literature was the evaluation criterion for UD facilities.

The studies in the second round were classified into 24 domestic and 64 international studies, and the researchers reviewed the suitability of the literature for the final analysis. We selected 14 studies: 9 related to UD facilities in Korea [16,23,24,25,26,27,28,29,30], 3 related to Korea’s inclusive design sports facilities [31,32,33], and 2 concerning overseas facilities based on UD and inclusive design [12,34]. The research team carefully read the collected literature and examined the content judged to be relevant to this study.

#### 2.2.3. Delphi Technique

First Delphi survey

The first Delphi survey allowed the panel to freely describe their answers to a subjective question that enquired about the factors to be considered when designing a sports facility for people with disabilities considering the UD. In addition, the factors to be applied first to the seven UD principles were selected.

Second Delphi survey

The second Delphi survey was a multiple-choice questionnaire regarding the 50 sub-factors of the 13 criteria obtained when considering the UD for physical education facilities through the first Delphi. In addition, for the 50 sub-factors, elements to be added or deleted were indicated. Mean, standard deviation, convergence, agreement, content validity, and reliability were analyzed.

#### 2.2.4. Expert Interviews

Criteria for interpreting the convergence, mean, standard deviation, agreement, content validity, and reliability values obtained through the Delphi surveys were established. Whether to converge, modify, or delete each sub-factor was discussed (Table 2).

#### 2.2.5. Data Analysis

The collected data were analyzed using mean, standard deviation, median, minimum, maximum, interquartile range, concordance, convergence, and content validity using SPSS for Windows version 12.0 and Excel. The data were analyzed by criterion, through intraclass correlation coefficients (ICCs). The benchmark mean and standard deviation were set to <3.50 and 1.0, respectively [35]. The degree of agreement and convergence level were set at 0.75 or higher and 0.50 or higher, respectively [36]. The content validity standard was set to 0.37 or higher, since the number of Delphi panelists was 21. The ICC suggested relatively stable values when the number of samples was small [37]. In general, the reliability index of the ICC was judged to be very high, relatively high, moderately high, and reasonable if it was 0.80 or more, 0.60 or more, 0.40 or more, and 0.20 or more, respectively [38].

## 3. Results

### 3.1. Literature Search Results

Nine studies related to UD facilities in Korea [16,23,24,25,26,27,28,29,30], three studies related to Korea’s inclusive-design sports facilities [31,32,33], two studies concerning overseas UD and inclusive design [12,34], and the elements of the evaluation indicators, such as background and necessity, goals, scope, and principles of the introduction of UD facilities, were reviewed. The literature review results are shown in Table 3.

### 3.2. Delphi Survey Results

#### 3.2.1. First Delphi Survey

Through the first Delphi survey, the elements to consider for the evaluation index for sports facilities for people with disabilities considering a UD were set. Thirteen criteria (gymnasium, accessibility, gym finishing material, stairs, elevator, corridor, lobby, entrance, reception desk, toilet, shower changing room, ancillary facilities, and common facilities finishing material) and 50 sub-factors were derived. The criteria were primarily classified into “living space” and “common space.” The living room space was defined as “factors of movement,” and the common space was defined as “movement and passage” and “incidental service.” The detailed results are shown in Table 4.

#### 3.2.2. Second Delphi Survey

The second Delphi survey confirmed the criteria and sub-factors of the evaluation indicators by examining the composition of the evaluation index for physical education facilities for people with disabilities, considering the UD derived from the first Delphi survey, using a five-point Likert scale.

The mean values and standard deviation values of the indicators were confirmed. When the five-point Likert scale was used in most previous studies, items with an average value of <3.50 and a standard deviation >1.00 were removed. The values of the mean and standard deviation were converged for all sub-factors. Second, the degree of convergence and the agreement were analyzed to refine the sub-factors. For the third-order item refinement, the content validity was less than the standard values for only one sub-factor item (no. 9). In the reliability verification, the index of the ICC was confirmed. The value of the ICC result was reliable, as for all items, it was 0.20 or higher. Table 5 provides the details showing the suitability of the evaluation index according to the Delphi survey.

### 3.3. Results of the Expert Meeting

An expert meeting was held to confirm the contents of the evaluation index for sports facilities for people with disabilities considering the UD. The sub-factors of the guard line were evaluated according to the convergence criteria set by the panel.

One sub-factor fell short of the standard and was deleted. Therefore, as shown in Table 5, an evaluation index based on 13 criteria and 49 sub-factors for sports facilities for people with disabilities considering the UD was finally confirmed.

## 4. Discussion

This study developed an evaluation index for sports facilities for people with disabilities considering the UD. Opinions on what factors should be considered when evaluating sports facilities based on the seven UD principles were collected and analyzed through an expert panel via the Delphi method.

The results revealed 3 median criterion factors and 13 lower criterion factors to consider for an evaluation index for sports facilities to which the UD is applied. Specifically, important reference elements are first, a living space for exercise that can be accessed safely and easily; second, a common space for the purpose of movement and passage that can safely provide convenience of movement; and, third, a common space for the purpose of promoting incidental services and social functions.

Space classification according to the general building law includes living space for a certain purpose and common space for temporary use [39]. According to these classification criteria, this study classified three median criteria—factors of movement, movement and passage, and incidental service—and further derived 13 sub-criteria.

First, the main reference factor living space is a space used for the original purpose of exercise owing to the nature of the gym, and the lower reference factors were classified into gymnasium and accessibility. Sports and leisure facilities should be designed to support and encourage the participation of people with mobility and disabilities in physical and social activities [40]. In particular, in the case of school sports facilities, the effectiveness of physical education should be increased through a redesign based on the UD, so that young students can access and use them more easily [41]. To emphasize the social value of sports and leisure programs that can strengthen social solidarity [42], the local residents’ use of public sports facilities should be promoted.

Owing to the nature of the living room space for exercise, sports facilities should be planned considering the users’ body types, physical strength, audience viewership, spatial diversification, safety, and convenience. Older adults’ health status and exercise ability should be considered. In addition, older adults’ vision requires consideration. The deterioration of older adults’ color discrimination ability hinders their ability to accurately perceive space, shape, and distance [43]. Thus, the wall colors, elevator locations, and format of information boards are important.

In contrast, difficulty in access to facilities is an environmental barrier to participation in sports [44]. A study reviewed physical activity restrictions perceived by children with disabilities and reported that inappropriate facilities and access limitations were barriers. Simultaneously, the accessibility of facilities should be improved to promote participation [45]. In particular, according to the UD principle “Size and Space for Access and Use,” non-disabled people and wheelchair users should have a comfortable access to any place. Furthermore, accessibility to facilities should be considered to make access easier for women wearing skirts or for small users [46]. In addition, the facility design to increase accessibility, such as those designs reported to have a significant effect on physical activity behavior [47], can have a positive influence on the use of public sports facilities as well as on the health of participating individuals.

Second, public spaces were classified into five reference sub-factors under movement and passage: stairs, lifts, corridors, lobby, and entrance doors. It is possible to reduce unnecessary movement and reduce the effort required, thereby enhancing the effectiveness of exercise for those who visit gyms.

Lobbies and corridors often contain slippery marble and granite finishes. Hence, it is necessary to use materials so to improve users’ stability. In particular, it was reported that it was necessary to consider the width of facilities and the presence of safety handles [48]. To prevent stair accidents, anti-slip materials must be considered.

Safety-related objects, such as handles to grip to avoid falls, should be easy to operate regardless of users’ age, knowledge, or ability. They should not cause unnecessary burdens on the user. Furthermore, their use should be simple and intuitive [49]. In addition, the lobby should be easy to use for everyone. Hence, the access distance from other spaces should be short, and all locations should be clearly marked [50].

Ultimately, for all users, including wheelchair users, people using crutches, and older adults with reduced mobility, the presence of non-skid floors and the installation of identifiable signs, pathways, and handrails will help prevent falls. Gyms need to be designed so they can be used by people of all ages and sexes, rather than by excessively considering their interior design. In particular, since stairs, passages, lobbies, and elevators are common spaces for movement, it is important to choose a design that can minimize the occurrence of minor safety problems and provide convenience.

Third, common space was classified into five sub-factors: reception desk, toilet, shower and changing room, ancillary facilities, and closing of public facilities under ancillary services. Ancillary facilities are additional facilities that satisfy various needs by providing convenient further services to the users in addition to the essential exercise service of public sports facilities. From this viewpoint, ancillary facilities within a public gym can be used as a space to promote social activities. In particular, interaction between users can promote social cohesion [20]. A study [51] reported that shower and convenience facilities should be expanded in sports facilities for people with disabilities. In addition, the necessity of ancillary service facilities in public spaces was also mentioned.

The reason the space for ancillary services in public sports facilities is increasing is that these supplementary service facilities are gradually expanding to meet the needs of users. In contrast, it is generally difficult to safely use existing outdoor playground facilities, as they do not provide countermeasure against physical danger. This was shown to be true also for older adults who cannot move for a relatively long time since these facilities do not offer any rest space [52].

Public sports facilities applying the UD principles will be suitable for all people. They will allow fair use and flexibility of use, provide recognizable information and simple and intuitive space, demand a low physical effort, and offer a suitable space, ensuring increasing accessibility and use [20].

This study has some limitations. The scope of the research was limited, and the research method lacked objectivity. Since the opinions of 13 experts were analyzed using the Delphi research method, it is difficult to generalize the results. Therefore, it is necessary to validate the evaluation index. However, the main strength of this study is the application of the seven UD principles and a proposal of how they can be used to promote social integration and equity.

## 5. Conclusions

This study was conducted to develop sub-factors and define criteria for an evaluation index of sports facilities for people with disabilities considering the UD, through an expert panel via the Delphi method. The criteria for the evaluation index developed through a literature review and Delphi surveys were, first, “factors of movement” and “living space”, which consisted of “gymnasium”, “accessibility”, and “gym finishing material”. Furthermore, “common space movement and passage” consisted of “stairs”. “elevator”, “corridor”, “lobby”, and “entrance”. “Incidental services of common space” consisted of “reception desk”, “toilet”, “shower changing room”, “ancillary facilities”, and “common facilities’ finishing material”. Hence, 49 sub-factors were derived.

The recommendations of this study are as follows. First, the developed evaluation index should be applied by the operators and users of public sports facilities used by both people with disabilities and those who are unemployed, and validation work should be conducted. Second, as a follow-up, guidelines for applying the UD to various sports facilities for people with disabilities should be developed. Third, it is necessary to discuss financial-related matters concerning the installation of facility equipment. Lastly, awareness education programs on whether UD sports facilities should be built and used by everyone without inconvenience should be developed.

## Figures and Tables

**Figure 1 healthcare-10-02151-f001:**
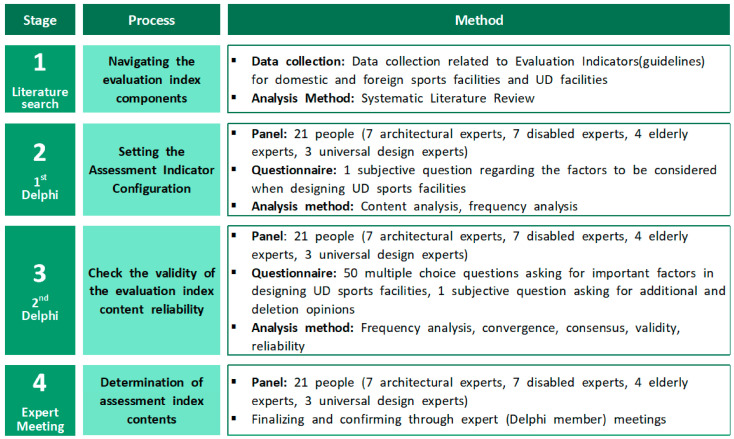
Research procedure.

**Table 1 healthcare-10-02151-t001:** Personal characteristics of the expert panel.

Career Field	Education	Career Period (years)	Age (years)
City planner	Ph.D.	28	51
Architectural engineer	Master’s	30	53
Architectural engineer	Master’s	30	55
Architect	Master’s	15	55
Architect	Master’s	10	38
Architect	Master’s	10	39
Professor of building environment	Master’s	10	39
Professor of special education	Master’s	10	57
Professor of special education	Master’s	12	48
Professor of special education	Master’s	13	46
Parasports facility operator	Ph.D.	13	45
Parasports facility operator	Ph.D.	14	39
Professor of welfare for persons with disabilities	Ph.D.	30	58
Professor of welfare for persons with disabilities	Master’s	13	44
Professor of welfare for the elderly	Master’s	27	52
Professor of welfare for the elderly	Master’s	13	38
Professor of welfare for the elderly	Ph.D.	18	46
Professor of welfare for the elderly	Master’s	17	47
Professor of universal design	Ph.D.	35	60
Professor of universal design	Ph.D.	10	56
Professor of universal design	Ph.D.	33	55

**Table 2 healthcare-10-02151-t002:** Evaluation indicators for determining the sub-factors.

Division	Standard
Convergence	When the mean, standard deviation, convergence, agreement, content validity, and reliability values are above the standard values
	If one of the values of mean, standard deviation, convergence, agreement, content validity, and reliability is below the standard value, convergence is achieved when more than 80% of the panel agree to convergence
Delete	When two or more values of mean, standard deviation, convergence, agreement, content validity, and reliability are below the standard values

**Table 3 healthcare-10-02151-t003:** Literature search results.

Area	Contents
Background	In aging societies, UD concerns the environment, information, and services that can be used by everyone UD can create a building that can be used by everyone without inconveniences through a design that embraces diversity
Necessity	Ensure that people of all abilities can use the space safely and conveniently without assistance
Goal	The needs of users should be met to provide fair participation and enjoyment
Range	UD building scope consists of access, entry, movement, and sanitary space
Principle	Provide a safe and convenient building environment for everyone to use Design that reflects people’s needs, continuing to improve, so that it is not at the legal minimum Reflect the seven UD principles as much as possible.

**Table 4 healthcare-10-02151-t004:** Criteria and sub-factors of the first Delphi survey.

Criteria			Subfactor	
Living space	Factors of movement	Gymnasium	1	Plan a variable space to play a variety of sports
			2	Space for installation of exclusive events considering body type and physical strength
			3	Stage and spectator seats are designed to be movable to secure the exercise space
			4	Plan for safety devices such as indoor fences
			5	Plan at least two evacuation routes in case of emergency
			6	Plan for a location adjacent to a ramp or lift
		Accessibility	7	Plan equally for all users to enter the gym
			8	Installation of information boards for easy access to internal facilities
			9	Provision of a device to provide program information
		Gym finishing material	10	Planning for anti-slip and shock resistance floor material
			11	Differentiate the color of the walls and elevators at each floor
			12	Installation of cushions and kick plates on walls, corners, etc.
Common space	Movement and passage	Stairs	13	Identifiable non-slip and finished material at the ends
			14	Emergency bell installation in stairwell
			15	Planned low and wide, with equal spacing of steps
			16	Continuous installation of handrails in both directions of the stairs and display of braille at the end
		Elevator	17	Install one or more large elevators that can be used by wheelchairs
			18	In case of a large scale, install elevators in each major area
			19	Finish to prevent collision of electric wheelchair
			20	Installation of handrails in elevators
		Corridor	21	Plan so that there are no obstacles; warning signs if obstacles are unavoidable
			22	Continuous installation of handrails in both directions of the hallway
			23	Plan wide enough for wheelchair access
			24	Braille guide board with position indication is attached on the wall of the room
		Lobby	25	Plan to make it easy to find
			26	Installation of a comprehensive guide map for information on the entire facility
			27	Secure enough space, plan a rest area
			28	Separate space plan for guide dogs
		Entrance	29	Prevention of safety accidents by controlling the opening and closing time of automatic doors
			30	Ensuring an effective width that allows wheelchair access
			31	Wheelchair cross-access for larger rooms
			32	Application of a glass that can be recognized by the users
			33	Handle plan that can be used equally by all users
	Incidental service	Reception desk	34	Plan in a recognizable location in the lobby
			35	Plan so that it can be used equally by all users
			36	Plan to enable the acquisition of various information on the facility
		Toilet	37	Use of anti-slip tiles and installation of handrails for disabled users
			38	Family toilet plan for severely disabled users
			39	Securing effective space for wheelchair rotation
			40	The sink is height-adjusted, and the mirror is angle-adjustable
		Shower changing room	41	Use of anti-slip tiles and installation of handrails for disabled users
			42	Family shower/changing room plan for people with severe disabilities
			43	Flat installation for changing clothes in the changing room
			44	Shower/changing room plan that can be used equally by all users
		Ancillary facilities	45	Exercise wheelchair storage space and location planned
			46	A separate rest area for gym users is planned
		Common facilities finishing material	47	Plan so that there is no difference when using the gym
			48	For walls, consider finishing to avoid crashes
			49	Installation of handrails to support users as they walk
			50	Consideration of durability to withstand non-slip material, wheelchair, etc.

**Table 5 healthcare-10-02151-t005:** Criteria and sub-factors of the Delphi evaluation index.

Criteria			Sub-Factor	Mean	SD	Median	Percentile		Convergence	Agreement	CVR	ICC
							25%	75%				
Living space	Factors of movement	Gymnasium	1	4.3811	0.805	5.000	4.000	5.000	0.500	0.800	0.619	0.699
			2	4.143	0.910	4.000	3.500	5.000	0.750	0.625	0.524	
			3	3.905	0.831	4.000	4.000	4.000	0.000	1.000	0.429	
			4	4.571	0.507	5.000	4.000	5.000	0.500	0.800	0.905	
			5	4.762	0.539	5.000	5.000	5.000	0.000	1.000	0.905	
			6	4.381	0.590	4.000	4.000	5.000	0.500	0.750	0.810	
		Accessibility	7	4.619	0.590	5.000	4.000	5.000	0.500	0.800	0.905	0.743
			8	4.476	0.680	5.000	4.000	5.000	0.500	0.800	0.714	
			9	3.857	0.910	4.000	3.000	5.000	1.000	0.500	0.143	
		Gym finishing material	10	4.667	0.577	5.000	4.000	5.000	0.500	0.800	1.000	0.626
			11	4.095	0.768	4.000	3.500	5.000	0.750	0.625	0.714	
			12	4.524	0.602	5.000	4.000	5.000	0.500	0.800	1.000	
	Movement and passage	Stairs	13	4.762	0.436	5.000	4.500	5.000	0.250	0.900	1.000	0.562
			14	4.333	0.658	4.000	4.000	5.000	0.500	0.750	0.810	
			15	4.429	0.598	4.000	4.000	5.000	0.500	0.750	0.905	
			16	4.810	0.402	5.000	5.000	5.000	0.000	1.000	1.000	
		Elevator	17	4.762	0.436	5.000	4.500	5.000	0.250	0.900	1.000	0.651
			18	4.571	0.507	5.000	4.000	5.000	0.500	0.800	1.000	
			19	4.524	0.602	5.000	4.000	5.000	0.500	0.800	0.905	
			20	4.810	0.512	5.000	5.000	5.000	0.000	1.000	0.905	
		Corridor	21	4.667	0.483	5.000	4.000	5.000	0.500	0.800	1.000	0.737
			22	4.571	0.507	5.000	4.000	5.000	0.500	0.800	1.000	
			23	4.714	0.463	5.000	4.000	5.000	0.500	0.800	1.000	
			24	4.762	0.436	5.000	4.500	5.000	0.250	0.900	1.000	
		Lobby	25	4.667	0.483	5.000	4.000	5.000	0.500	0.800	1.000	0.685
			26	4.619	0.669	5.000	4.000	5.000	0.500	0.800	0.810	
			27	4.476	0.680	5.000	4.000	5.000	0.500	0.800	0.810	
			28	4.476	0.512	4.000	4.000	5.000	0.500	0.750	1.000	
		Entrance	29	4.667	0.577	5.000	4.000	5.000	0.500	0.800	0.905	0.769
			30	4.762	0.436	5.000	4.500	5.000	0.250	0.900	1.000	
			31	4.429	0.507	4.000	4.000	5.000	0.500	0.750	1.000	
			32	4.476	0.602	5.000	4.000	5.000	0.500	0.800	0.905	
			33	4.667	0.483	5.000	4.000	5.000	0.500	0.800	1.000	
	Incidental service	Reception desk	34	4.571	0.507	5.000	4.000	5.000	0.500	0.800	1.000	0.685
			35	4.714	0.561	5.000	4.500	5.000	0.250	0.900	0.905	
			36	4.571	0.507	5.000	4.000	5.000	0.500	0.800	1.000	
		Toilet	37	4.857	0.359	5.000	5.000	5.000	0.000	1.000	1.000	0.459
			38	4.714	0.561	5.000	4.500	5.000	0.250	0.900	0.905	
			39	4.857	0.359	5.000	5.000	5.000	0.000	1.000	1.000	
			40	4.667	0.483	5.000	4.000	5.000	0.500	0.800	1.000	
		Shower changing room	41	4.810	0.402	5.000	5.000	5.000	0.000	1.000	1.000	0.699
			42	4.619	0.498	5.000	4.000	5.000	0.500	0.800	1.000	
			43	4.476	0.602	5.000	4.000	5.000	0.500	0.800	0.905	
			44	4.714	0.561	5.000	4.500	5.000	0.250	0.900	0.905	
		Ancillary facilities	45	4.238	0.700	4.000	4.000	5.000	0.500	0.750	0.714	0.554
			46	4.095	0.831	4.000	4.000	5.000	0.500	0.750	0.619	
		Common facilities finishing material	47	4.714	0.561	5.000	4.500	5.000	0.250	0.900	0.905	0.786
			48	4.619	0.590	5.000	4.000	5.000	0.500	0.800	0.905	
			49	4.190	0.750	4.000	4.000	5.000	0.500	0.750	0.524	
			50	4.619	0.590	5.000	4.000	5.000	0.500	0.800	0.905	

Note: The sub-factors meanings are the same as those of the sub-factors in Table 4.

## Data Availability

The data presented in this study are available on request from the author.

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
