# Peer review of "Development of Evaluation Indicators for Sports Facilities for People with Disabilities Considering the Universal Design: Focusing on the Republic of Korea"

_healthcare, 2022, doi:10.3390/healthcare10112151_

Round 1
Reviewer 1 Report
Thank you for the opportunity to review this manuscript on UD evaluation indicators for sport facilities for people with disabilities. This research is important to advancing opportunities for all individuals. This paper was well written and the methodology is sound. I recommend accepting it with only one minor request. On page 11, lines 295-299, the information on vision does not align with the preceding information. It is unclear where the authors are trying to go with these statements. Expand further explaining the practical relevance of this issue.
Author Response
"첨부파일을 봐주세요."

Reviewer 2 Report
The article deals with a topic of interest. I would like to congratulate the authors.
Some points that need to be improved have been identified.
INTRODUCTION:
When mentioning disabilities in the introduction, it would be useful to detail what kind of disabilities. E.g. lines 42, 53, 56
The 7 principles of UD (LINE 129) should be explained or announced in a generic way in the theoretical framework.
According to the actions described in the methodology, the objective should be improved.
Conduct a systematic review of the literature on Sports facilities for people with kinetic disabilities...
2. Introduce the other objective described in the article. line 121 in the objective should better specify what type of disabilities this study considers, visual, locomotor, intellectual...
MATERIALS AND METHODS
Design
First of all, it should be indicated that the study corresponds to several studies.
1. Systematic review. It should be justified that a systematic review has actually been carried out. There are doubts, in the light of what the authors describe.
Figure 1 indicates that the first literature search stage consisted of a Systematic Literature Review. Later, in section 2.2.1, lines 144 and 149, some information is shown. However, the systematic review, a step prior to possible meta-analysis studies, requires rigorous methodological procedures that are not described in this article. It should be explained precisely.
2. Mixed methods. Use of qualitative (expert interview, content analysis) and quantitative (frequency analysis) instruments and analysis of results.
Participants (panel experts):
Insert a section referring to the participants. Table 1 should be placed here.
Their characteristics should be described.
It would also be convenient to introduce the authorisation of an official ethics committee and to put the corresponding code.
Instruments and Procedure
Systematic review. Here the procedure followed should be described very exhaustively. If it is a systematic review, all the details of this study should be given (authors are advised to review the literature on this type of study).
Delphi technique. Include here the description made by the authors. The UD's seven principles can be mentioned.
Expert interview. Include this section here as it is the place where all the strategies used are described.
Data analysis.
Review the statistical strategies used and the way they are described. This section should be improved.
The rest of the sections, corresponding to results, discussion and conclusions, should be ordered according to the objectives.
Author Response
"Please see the attachment."

Reviewer 3 Report
The authors aimed to develop evaluation indicators by selecting items that comprehensively evaluated sports facilities for people with disabilities considering a universal design by conducting Delphi surveys centered on such sports facilities and sports experts.
The article content is attractive but has several flaws. Also, extensive editing of the English language and style is must required.
Introduction
Lines 40-42: How do authors mean with "..major social risk society". It seems to be not the most appropriate term.
Lines 63-65: Please explain these concepts to the readers
Lines 115-118: This paragraph is difficult to follow. Indeed, the introduction has no good readability. Paragraphs and sentences should be shorter.
Lines 119-122: The study's purpose is difficult to follow. I suggest that the authors rewrite this sentence.
Materials and methods
Lines 144-149: Which guideline is followed for the systematic review?
Results
Table 5 has no labels and notes. In the current version, the sub-factors meanings are missing. Please consider reformulating this table.
Discussion
Lines 260-268. The authors should provide the main findings of the study in this paragraph. Moreover, the discussion is too long and wordy. I suggest a reorganization of concepts and ideas discussed here.
Author Response
"Please see the attachment."

Round 2
Reviewer 2 Report
After reviewing the responses to my comments from the first review, I consider that this version has substantially improved on the previous one. It could therefore be accepted for publication.
Author Response
우리 연구를 출판할 수 있도록 허락해주셔서 감사합니다.
Reviewer 3 Report
Although the effort from the authors the manuscript is very difficult to follow in the present form. The introduction has not a clear rationale, there are no sufficient systematic methods to support the rigor of a systematic review, and the discussion is still long and wordy. Lastly an extensive editing of English language is required.
Author Response
"첨부파일을 봐주세요."
